# High Molecular Weight Kininogen: A Review of the Structural Literature

**DOI:** 10.3390/ijms222413370

**Published:** 2021-12-13

**Authors:** Michał B. Ponczek

**Affiliations:** Department of General Biochemistry, Faculty of Biology and Environmental Protection, University of Lodz, Pomorska 141/143, 90-236 Lodz, Poland; michal.ponczek@biol.uni.lodz.pl; Tel.: +48-426354483

**Keywords:** kininogen, structure, AlphaFold, cryo-EM, proteins, kinin, bradykinin

## Abstract

Kininogens are multidomain glycoproteins found in the blood of most vertebrates. High molecular weight kininogen demonstrate both carrier and co-factor activity as part of the intrinsic pathway of coagulation, leading to thrombin generation. Kininogens are the source of the vasoactive nonapeptide bradykinin. To date, attempts to crystallize kininogen have failed, and very little is known about the shape of kininogen at an atomic level. New advancements in the field of cryo-electron microscopy (cryoEM) have enabled researchers to crack the structure of proteins that has been refractory to traditional crystallography techniques. High molecular weight kininogen is a good candidate for structural investigation by cryoEM. The goal of this review is to summarize the findings of kininogen structural studies.

## 1. Introduction

Kininogens or Fitzgerald factors are multidomain glycoproteins found in the blood of most vertebrates [1,2,3,4]. The orthological proteins (kininogen-1, UniProt ID: P01042 for human) are produced by splicing of the KNG1 gene transcript. High molecular weight kininogen (HK) is one of the members of the kallikrein-Kinin system (KKS). Together with coagulation factor (f)XI, the KKS form the contact activation system of blood coagulation (CAS) or more commonly known as the intrinsic pathway of coagulation contributing to thrombin generation primarily under pathologic conditions. CAS can operate independently from the tissue factor (TF) extrinsic pathway (Figure 1). As a class, kininogens have been studied for decades, nonetheless, the atomic arrangement of full length kininogen or its fragments are still unknown for all species with no coordinates in Protein Data Bank (PDB). A protein Basic Local Alignment Search Tool (BlastP) applied with human kininogen amino acid sequence against PDB returns only proteins containing homologous cystatin domains with identity around 30% (Table 1). 

Moreover, entering the keyword “kininogen” in the database search engine returns only structures containing polypeptide fragments referred to as “inserts of kininogen peptides”, “peptides derived from kininogen” or bradykinin (BK) derived peptides (Table 2). 

The purpose of this review article is to summarize the finding of structural studies that involve kininogens and emphasize how a good-quality spatial structure would allow for a better understanding of their role in disease and as therapeutic targets. 

## 2. HK Biochemical and Evolutionary Characterization

The human high molecular weight kininogen (huHK) is a 626 amino acid glycoprotein (devoid of signal peptide). The molecular mass is only 70 kDa, but because it undergoes numerous post-translational modifications, the real mass is much higher and it run ~120 KDa on SDS-PAGE under non-reducing conditions [1,2,3,4]. huHK circulates in plasma at a concentration of approximately 70 µg/mL. The protein is mostly expressed by hepatocytes and secreted into the blood. huHK theoretical isoelectric point based on amino acid sequence was calculated using Compute pI/Mw tool (https://web.expasy.org/compute_pi/, accessed on 29 October 2021) to be 6.23, however the experimental reported pI is 4.9 ± 0.2 because of high sialic acid content. The sialic acid constitutes almost 42% of HK molecular mass content. A highly purified human kininogen has been established to be about 8.6 mol/mol (50 kDa) [5]. This explains the discrepancy between the theoretical mass from the amino acid sequence and the experimental values of HK weight. 

In plasma, huHK circulates as non-covalent complex with PK [6,7,8]. Mouse plasma studies as well as huHK case reports demonstrated an important carrier function of HK where HK deficiency was accompanied with low PK levels [9]. Similar to its interaction with PK, huHK also forms non-covalent complex with fXI (a PK homolog, produced by duplication of the PK gene) [4,10,11,12,13,14]. There is enough huHK in the plasma to cover the entire range of PK and fXI. In contrast to PK, the carrier properties of HK are not required for maintaining fXI levels [10]. Kininogens, mostly HK, have many different biological activities, including: the role in blood coagulation by helping to locate suitably plasma prekallikrein (PK) and fXI next to fXII on negatively charged surfaces, inhibition of blood platelets aggregation induced by thrombin and plasmin, and the liberation of vasoactive peptide BK (Figure 1). BK manifests important physiological effects: smooth muscle contraction, induction of hypotension, natriuresis and diuresis, and decrease in blood glucose level. It is also a mediator of inflammation and causes and increase in vascular permeability, release of mediators of inflammation like prostaglandins, stimulation of nociceptors and has a direct and indirect cardioprotective effect [4].

HK is organized into distinct domains designated D1 through D6 that can be listed as follows with the amino acid positions numbering: D1: 1–113, D2: 114–234, D3: 235–357, D4: 358–383, D5: 384–502, and D6: 503–626. The domains 1, 2, 3, 5 and 6 are roughly 120 amino acids in length and Domain 4 is only 26 amino acids long. Domains 1 and 6 are connected by a single disulfide bridge involving Cys 10 and Cys 596 [4] (Figure 2). 

Some features and functions are recognized for these domains. Domains 1–3 belong to a family of cysteine protease inhibitors (Cystatins, MEROPS inhibitor family I25, clan IH) [6,7,8,9] and sequence homology can be simply searched based on human HK sequence using online databases of protein domains like PFAM (http://pfam.xfam.org/, accessed on 19 October 2021) [15,16,17,18,19,20,21,22,23,24], PROSITE (https://prosite.expasy.org/, accessed on 19 October 2021) [25,26,27,28,29,30,31,32,33,34,35] or SMART (http://smart.embl-heidelberg.de/, accessed on 19 October 2021) [36,37,38,39]. As cystatins-like domains their inhibitory action was confirmed experimentally with inhibition of atrial natriuretic factor by D1 and inhibition of calpain and papain (cysteine or thiol proteases) by D2 and D3, respectively. HK is cleaved at two positions in D4 (the smallest domain of HK) by plasma kallikrein (protease form of PK) to release the nonapeptide bradykinin (RPPGFSPFR) and the two-chain HKa. BK boosts vasodilation and vascular permeability and can develop soft tissue swelling that can be life threatening as that observed in hereditary angioedema [4]. 

D5 and 6 are vital for the contact pathway of blood coagulation. D5 is rich in histidine motifs facilitating binding to the negatively charged surfaces anchoring PK and fXI in close proximity to fXII. D6 contains overlapping stretches of amino acids that interacts in the complex formation with the coagulation fXI and PK [40,41,42,43]. 

Human HKa run as a two bands on SDS-PAGE under reducing conditions corresponding to a heavy, spanning amino acid residues 1–362 (D1–D3 and the first six amino acids of the short Domain 4), and a light chain consisted 255 amino acids (D5 and D6). huHK is heavily glycosylated with 3 N-linked carbohydrates on the heavy chain and 9 O-linked carbohydrates in the light chain of huHK. Many of the carbohydrate derivatives in huHK are sialic acid polymers [4,5]. Messenger RNA alternative splicing transcript of the KNG1 gene is translated into a smaller protein variant (70 kDa)–low molecular weight kininogen (LK). LK has identical domains 1–4 as huHK albeit Domain 5 is shortened by 38 amino acids in length and designated D5L while D6 is lacking. Digestion of LK by tissue kallikrein (TK) liberates a 10 amino acid peptide KRPPGFSPFR-Lysyl-Bradykinin (LBK) which is subsequently split to BK by the enzyme arginine aminopeptidase. LK is not involved in blood clotting but can still inhibit the aggregation of blood platelets. Domain interaction of HK has been reviewed extensively in Winter et al. (2020) publication [4]. 

Thanks to genome and transcriptome sequencing advancements, it is nowadays known that a similar situation with KNG1 gene messenger RNA alternatively splicing occurs not only in mammals, but also in reptiles, birds and amphibians, where HK and LK share D1 through D4 domains, but have varied D5 domains. D6 was found in HK from all tetrapods, suggesting HK carries PK in most terrestrial vertebrates. HK D5 is more mutable and in amphibians, crocodilians, birds, and turtles it is shorter than in mammals and is absent in lizards and snakes (Appendix A). Cartilaginous or ray-finned fish kininogens are more like mammalian LK than to HK where D5 or D6 domains are absent. Although kininogens are found in lobe-finned fishes, the coelacanth and the lungfish have a D6 domain with short histidine-rich motifs corresponding to a more D5 domain configuration. The appearance of HK in the lobe-finned fishes indicates an adaptation to blood clotting activation upon contact with silica-rich soil or mud impurities, coinciding with fXII appearance in the lungfishes, but fXII disappearance in birds and cetaceans where contact with the soil is much less frequent may indicate that contact activation is not very vital. Moreover, bird HK has shorter D5 domains (Domains 5 and 6 are important for the contact), but it seems it is still needed for kinin generation, as well as in cetaceans, where HK seems normal (Appendix A), however, fXII and PK are not expressed as corresponding DNA fragments are degraded, pseudogene and non-gene junk DNA fragment, respectively [13]. Evolutionary and functional links between orthological HK sequences can be enriched after solving the spatial structure for the human protein to provide an adequate template for homology modeling.

## 3. The Importance of Kininogen in Civilization and Infectious Diseases

HK significance, beyond physiological BK production, is participation in thrombotic disorders as it participates as cofactor for the contact activation system on many negatively charged surfaces (e.g., silica, nucleic acids, polyphosphates of different origins, misfolded proteins, collagen, bacteria, or viruses). Thrombosis and embolism are significant problems connected to civilization diseases related to the circulatory system where the role of HK should not be neglected. Thrombotic complications and coagulation disorders appear frequently secondary to infections with various pathogens where contact activation is triggered. BK released from HK during either contact activation or as part of the KKS contribute to the pathology of many inflammatory diseases like hereditary angioedema. Coronavirus SARS-CoV-2 the virus causing COVID-19 pandemic is known to bind Angiotensin-Converting Enzyme 2 (ACE2). Such binding increases BK release which promotes inflammation in the lungs, causing cough and fever, as well as further activation of the coagulation and the complement system. Emerging thrombotic complications associated with COVID-19 infection and reports of post-vaccination thrombotic reactions indicate the need to investigate the relationship within the hemostatic system, including contact factors [44,45]. Understanding the structure of HK is important for a better understanding of the changes and processes it undergoes not only in physiological state, but pathological situations like thrombotic disorders accompanying various civilization and infectious diseases, with the activation of the immune system, inflammation, and coagulation.

## 4. Literature Describing the Structure of Kininogen

Although images of huHK were made many years ago by transmission electron microscopy (TEM) using various negative staining techniques, they do not shed much light on the actual shape of the kininogen [3,13,14]. In the oldest work by Weisel et al. (1994) [3], where TEM images of human kininogen were taken, the samples were applied to a substrate from mica with tungsten shading and an 80 kV Philips 400 microscope was used to obtain pictures at 60,000× magnification. Using the scale provided by the authors, 3 domains of the kininogen seem to have a total of 130–180 Å, while one domain was 50–90 Å. In this work, together with HK images, IgG antibodies (mass ~150 kDa, length 100–150 Å) were visualized in some figures. Based on these data (Appendix A), it can be calculated that the length of the antibody is about 150 Å, and the kininogen associated with this antibody is ~180 Å long. Perhaps what the authors of this study showed were not individual kininogen molecules, but rather their aggregates, where a single domain is actually one kininogen molecule. 

In 2001, Herwald et al. (2001) [44] used 5 µL of sample with a concentration of 5–19 µg per mL in 50 mM Tris. HCl, pH 7.4 buffer plus 150 mM NaCl to carry negative stain imaging on a glow discharged carbon-coated copper grids using JEOL 1200 EX 60 kV transmission electron microscope. In this work, according to the scale given in some figures, the cation-free (EDTA treated) or Mg^2+^ treated protein molecules have estimated dimensions of 95 ± 30 Å (width) and 150 ± 30 Å (length) (Appendix A) with two larger domains (55 ± 20 Å) and three smaller domains (35 ± 20 Å) of 130–150 Å, and the substructure (subdomain) were visible. In the presence of Zn^2+^ ions, the protein was more spherical and compact, with the dimensions reduced to 90 ± 30 Å (width) and 110 ± 30 Å (length) (Appendix A).

In 2009, Oehmcke, Mörgelin, and Herwald reported TEM images of HK [45], and according to the image scale (Appendix A), the molecule should have only about 45 Å diameter. The internal structures corresponding to five to six domains were visible in those images. The dimensions of a given domain are only 15–20 Å. The authors did not describe the TEM methodology in the paper but referred to an earlier work by the group [44], and they did not provide numerical dimensions. In this study, the scale appears to underestimate the results and is not in agreement with previous reports.

In conclusion, based on the available literature with published TEM results of kininogen, there is no clarity and consistency as to the size of this molecule and its domains. This underscores the need for microscopic examination using more modern techniques than those from more than a decade ago.

The failure to publish the structure of HK may be due to the difficulty of its crystallization even in parts and inability to use the NMR technique, as the protein is too large. Unfortunately, all classic structure prediction methods, based on homology, threading, and de novo (ab initio), have failed. In the first two cases, it was due to the lack of appropriate, similar enough templates, and in the last, it was due to the huge size of this protein. Using new tools such as AlphaFold Protein Structure Database, a structure for HK by DeepMind calculations de novo (https://www.alphafold.ebi.ac.uk/entry/P01042, accessed on 19 October 2021) could be predicted (Figure 3) [46,47,48,49]. Unfortunately, despite the solution of cystatin domains (D1–D3) with good model confidence, the other domains of the protein have very low confidence and appear as almost unfolded floating chains, and no glycans were present (Figure 3).

A potential solution for this large 120 kDa protein containing many sugar residues is the use of cryogenic electron microscopy (cryo-EM). This will omit the bottle neck in traditional crystallography, as the formation of crystals is no longer needed. In contrast, a sample of low concentration protein solution is frozen to cryogenic temperatures by liquid ethane and fixed in an environment of thin film of vitreous ice to take electron microscope pictures of plenty protein molecules in different random orientations. Such flat two-dimensional images are classified and processed by powerful computers to generate a three-dimensional structure [52,53,54,55,56]. Nowadays, available cryogenic microscopes with sensitive detectors and appropriate software are available and enable the resolution of some proteins at 1.22 Å using a 300 kV beam of electrons [52,53,55]. More common and accessible dissemination of this technique should allow the solution of good-quality kininogen structures. 

## 5. Conclusions

Kininogens are important proteins playing a role in normal human physiology and pathology. Kininogens are part of the intrinsic coagulation system and the kallikrein–kinin systems and the source of the vasoactive nonapeptide BK. BK in turn has role in normal physiology as well as in disease. Despite the knowledge of the primary structure and the domain organization, still, there are no known spatial 3D structures of the whole proteins or its individual domains. Developed and improved visualization techniques providing near and/or atomic-level resolution provide the light at the end of the tunnel to resolve these structures. The application of cryo-EM to determine the spatial structure of kininogens will draw new frontiers in understanding the function of these proteins and open new pathways for drug development.

## Figures and Tables

**Figure 1 ijms-22-13370-f001:**
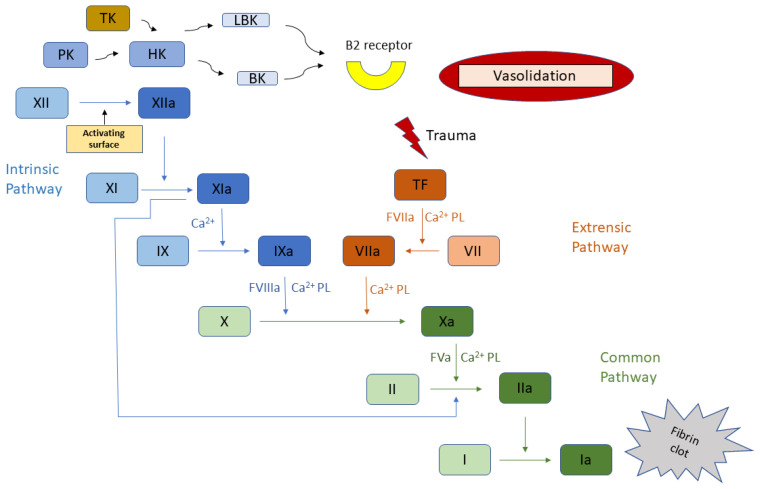
Blood coagulation cascade. Roman numerals denote individual coagulation factors, the letter “a” stands for the active form, other abbreviations are as in the text.

**Figure 2 ijms-22-13370-f002:**
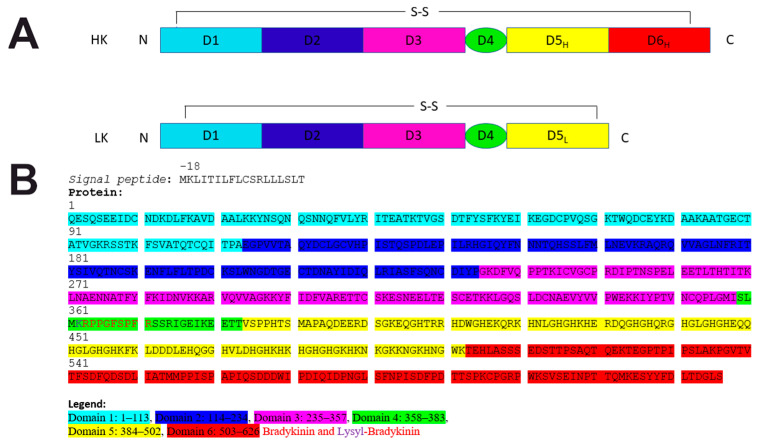
(**A**) Human plasma kininogen showing alternative splicing products of KNG1 gene messenger RNA–HK and LK. Both forms of protein have similar D1, D2, D3, and D4 domains, but different D5 domains. Domain 6 is absent in LK and the molecular weight of this form is lower than that of HK. (**B**) Human HK amino acid sequence according to UniProt P01042 with the indication of domains. N-terminus and C-terminus marked by corresponding letters. Disulfide (SS) bonds between the respective domains are marked.

**Figure 3 ijms-22-13370-f003:**
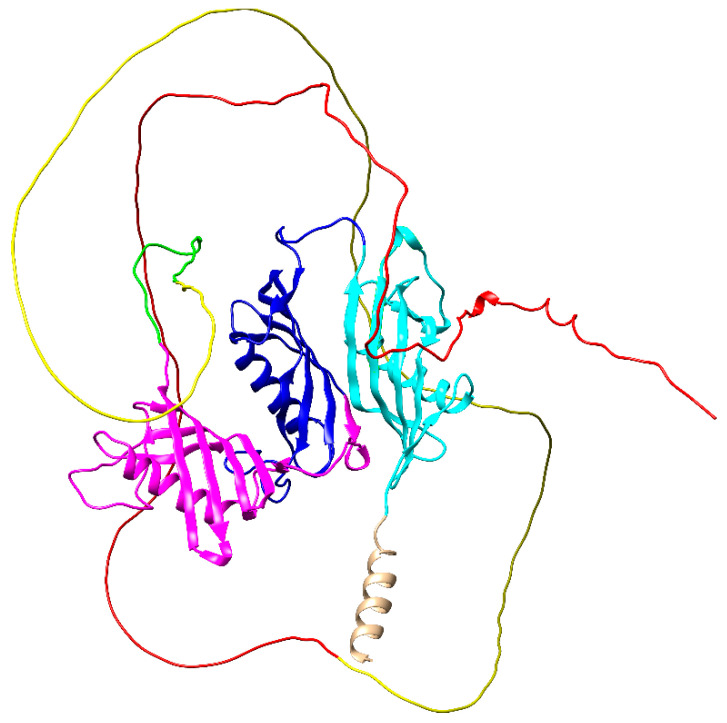
AlphaFold prediction of human HK. Colors of domains according to Figure 2. The structure contains a signal peptide colored brown. The figure was prepared in UCSF Chimera 1.15 based on downloaded AlPhaFold PDB coordinates (accessed date 29 October 2021) [50,51].

**Table 1 ijms-22-13370-t001:** Blastp results of human HK amino acid sequence against Protein Data Bank proteins.

Description	Percent Identity ^1^	PDB Accession and Chain ^2^
Cleaved human fetuin-b in complex with crayfish astacin [*Homo sapiens*]	26.20	6SAZ B
Crystal structure of L68V mutant of human cystatin C [*Homo sapiens*]	29.73	3PS8 A
Human Cystatin C; Dimeric Form With 3d Domain Swapping [*Homo sapiens*]	30.63	4N6M A
Crystal structure of human cystatin E/M [*Homo sapiens*]	29.73	1G96 A
Crystal structure of monomeric human cystatin C stabilized against aggregation [*Homo sapiens*]	31.48	4N6L A
N-Truncated Human Cystatin C; Dimeric Form With 3D Domain Swapping [*Homo sapiens*]	28.57	3GAX A
Crystal structure of V57G mutant of human cystatin C [*Homo sapiens*]	27.84	1R4C A
Crystal structure of V57P mutant of human cystatin C [*Homo sapiens*]	28.83	6ROA A
Hinge-loop mutation can be used to control 3D domain swapping and amyloidogenesis of human cystatin C [*Homo sapiens*]	28.83	3S67 A

^1^ Percent identity is less than 30%. Cystatin domain homology is detected. None is HK fragment. ^2^ PDB according to https://www.rcsb.org/, https://www.ebi.ac.uk/pdbe/ and https://pdbj.org/, accessed on 29 October 2021.

**Table 2 ijms-22-13370-t002:** The best ten of 39 hits for searching “kininogen” keyword against RCSB PDB: https://www.rcsb.org/, accessed on 29 October 2021.

ID	Title	Released
4ECC	Chimeric GST Containing Inserts of Kininogen Peptides	16 May 2012
4ECB	Chimeric GST Containing Inserts of Kininogen Peptides	16 May 2012
6BFP	Bovine trypsin bound to potent inhibitor	31 October 2018
4JD9	Contact pathway inhibitor from a sand fly	16 October 2013
6F3Y	Backbone structure of Des-Arg10-Kallidin (DAKD) peptide bound to human Bradykinin 1 Receptor (B1R) determined by DNP-enhanced MAS SSNMR	10 January 2018
6F3X	Backbone structure of Des-Arg10-Kallidin (DAKD) peptide in frozen DDM/CHS detergent micelle solution determined by DNP-enhanced MAS SSNMR	10 January 2018
6F3W	Backbone structure of free bradykinin (BK) in DDM/CHS detergent micelle determined by MAS SSNMR	10 January 2018
6O1S	Structure of human plasma kallikrein protease domain with inhibitor	6 March 2019
6O1G	Full length human plasma kallikrein with inhibitor	6 March 2019
6F27	NMR solution structure of non-bound [des-Arg10]-kallidin (DAKD)	10 January 2018

## Data Availability

Not applicable.

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
