# Peer review of "High Molecular Weight Kininogen: A Review of the Structural Literature"

_ijms, 2021, doi:10.3390/ijms222413370_

Round 1

Reviewer 1 Report

it would be valuable to supplement the article with the HK structure prediction results using other than AlphaFold methods.

Author Response

We would like to thank the anonymous reviewers, whose comments are very valuable to improve the quality of the final manuscript and have included the answers below:

Review Report (Reviewer 1)

“it would be valuable to supplement the article with the HK structure prediction results using other than AlphaFold methods.”

The manuscript was updated with sentence: “Unfortunately, all classic structure prediction methods, based on homology, threading and de novo (ab initio), have failed. In the first two cases due to the lack of appropriate, similar enough templates, and in the last due to the huge size of this protein.”

We have checked Swiss Model (homology modelling server - https://swissmodel.expasy.org/) and  Zang Lab (https://zhanggroup.org/) I-TASSER (Iterative Threading ASSEmbly Refinement,) but the obtained results are not worth mentioning in details as available the templates have very little similarity to HK. De novo methods like Zang Lab Quark are unsuitable due to the size of the protein and the numerous modifications with the residues of unknown sugar derivatives (possibly sialic acid).

Reviewer 2 Report

I have reviewed the manuscript “What we know about kininogen structure – importance for function and perspectives from cryo-EM” and found this work within the scope of IJMS. The authors reviewed the Kininogens multidomain glycoproteins that are found in the blood of most vertebrates. The review is very superficial, and not reporting comprehensive information. Title is not clear; such as cryo-EM should be defined. Abstract is giving a very brief information, without giving an idea about what the reader should aspect in the main body.

Figure 1 quality is very poor and hard to read.

Line 166,  181-84: paper Fig. 9B, C not a suitable format as the reader is not likely to go back to these papers to search paper for Fig. 9B, C. Therefore, the author should either add these figures with copyright permission or modified the related information with citations only.

There is some repetition of the information, please make sure to reduce the repetition.

Author Response

We would like to thank the anonymous reviewers, whose comments are very valuable to improve the quality of the final manuscript and have included the answers below:

Review Report (Reviewer 2)

“I have reviewed the manuscript “What we know about kininogen structure – importance for function and perspectives from cryo-EM” and found this work within the scope of IJMS. The authors reviewed the Kininogens multidomain glycoproteins that are found in the blood of most vertebrates. The review is very superficial, and not reporting comprehensive information.”

The review was completely redrafted and reedited updated with additional data and literature references to report more comprehensive information.

“Title is not clear; such as cryo-EM should be defined. Abstract is giving a very brief information, without giving an idea about what the reader should aspect in the main body.”

The title was changed “High Molecular Weight Kininogen: A review of the structural literature”. The Abstract was also updated.

“Figure 1 quality is very poor and hard to read.”

Figure 1 in graphical format quality is in agreement with MDPI requirements, the pasted image in the manuscript was rescaled to be larger and better visible for the reader. We apologize that the figure 1 was scaled too small in the first version of manuscript.

“Line 166,  181-84: paper Fig. 9B, C not a suitable format as the reader is not likely to go back to these papers to search paper for Fig. 9B, C. Therefore, the author should either add these figures with copyright permission or modified the related information with citations only.”

The information about previous papers together with copies of figures with appropriate redistribution rights for each journal and publishing house were implemented in Supplementary materials figures 2-4 :

reproduction of figure 3 (a): Weisel, J.W.; Nagaswami, C.; Woodhead, J.L.; dela Cadena, R.A.; Page, J.D.; Colman, R.W. The Shape of High Molecular Weight Kininogen. Organization into Structural Domains, Changes with Activation, and Interactions with Prekallikrein, as Determined by Electron Microscopy. Journal of Biological Chemistry 1994, 269, doi:10.1016/s0021-9258(17)36995-8.) The reproduction is presented according to JBC CC-BY license  https://www.asbmb.org/journals-news/copyright-and-reproduction.

reproduction of figure 9 A-D: Herwald, H.; Mörgelin, M.; Svensson, H.G.; Sjöbring, U. Zinc-Dependent Conformational Changes in Domain D5 of High Molecular Mass Kininogen Modulate Contact Activation. European Journal of Biochemistry 2001, 268, doi:10.1046/j.1432-1033.2001.01888.x.). The reproduction is presented according to FEBS letters Creative Commons License.

reproduction of figure 1a: Oehmcke, S.; Mörgelin, M.; Herwald, H. Activation of the Human Contact System on Neutrophil Extracellular Traps. Journal of Innate Immunity 2009, 1, doi:10.1159/000203700. The reproduction is presented according to reuse permission of the figure as MDPI an Karger, are signatories of the STM permission guidelines.

“There is some repetition of the information, please make sure to reduce the repetition.”

The repetitions were reduced during reediting the manuscript.

Round 2

Reviewer 1 Report

I accept all changes

Reviewer 2 Report

Many thanks for revising the manuscript.